# Ethnicity and impact on the receipt of cognitive–behavioural therapy in people with psychosis or bipolar disorder: an English cohort study

Rohan Michael Morris [1,2,3] William Sellwood,[1] Dawn Edge,[4] Craig Colling,[5,6] Robert Stewart,[5,6] Caroline Cupitt,[6] Jayati Das-Munshi[7]

► Prepublication history and additional materials for this paper is available online. To view these files, please visit the journal online (http://dx.doi.org/10.1136/bmjopen-2019-034913).

For numbered affiliations see end of article.

**Correspondence to**
Dr Rohan Michael Morris;
r.morris857@gmail.com

## ABSTRACT

**Objectives** (1) To explore the role of ethnicity in receiving cognitive–behavioural therapy (CBT) for people with psychosis or bipolar disorder while adjusting for differences in risk profiles and symptom severity. (2) To assess whether context of treatment (inpatient vs community) impacts on the relationship between ethnicity and access to CBT.

**Design** Cohort study of case register data from one catchment area (January 2007–July 2017).

**Setting** A large secondary care provider serving an ethnically diverse population in London.

**Participants** Data extracted for 30 497 records of people who had diagnoses of bipolar disorder (International Classification of Diseases (ICD) code F30-1) or psychosis (F20–F29 excluding F21). Exclusion criteria were: <15 years old, missing data and not self-defining as belonging to one of the larger ethnic groups. The sample (n=20 010) comprised the following ethnic groups: white British: n=10 393; Black Caribbean: n=5481; Black African: n=2817; Irish: n=570; and 'South Asian' people (consisting of Indian, Pakistani and Bangladeshi people): n=749.

**Outcome assessments** ORs for receipt of CBT (single session or full course) as determined via multivariable logistic regression analyses.

**Results** In models adjusted for risk and severity variables, in comparison with White British people; Black African people were less likely to receive a single session of CBT (OR 0.73, 95% CI 0.66 to 0.82, p<0.001); Black Caribbean people were less likely to receive a minimum of 16-sessions of CBT (OR 0.83, 95% CI 0.71 to 0.98, p=0.03); Black African and Black Caribbean people were significantly less likely to receive CBT while inpatients (respectively, OR 0.76, 95% CI 0.65 to 0.89, p=0.001; OR 0.83, 95% CI 0.73 to 0.94, p=0.003).

**Conclusions** This study highlights disparity in receipt of CBT from a large provider of secondary care in London for Black African and Caribbean people and that the context of therapy (inpatient vs community settings) has a relationship with disparity in access to treatment.

## Strengths and limitations of this study

► A key strength of this study is that the data were from a near-complete case register of a large secondary care mental health service provider, which has a near monopoly on mental health provision in its catchment area.

► Published data are available on the tools used for extracting information about cognitive–behavioural therapy, which indicates high degrees of precision (95%) and sensitivity (96%).

► A limitation of this study is that it was not possible to assess access to other types of psychological intervention (eg, family therapy).

► This study was not able to assess the offer of therapy (only receipt); consequently, it is unclear if there are ethnic differences in whether therapy is offered to Black service users.

## INTRODUCTION

### Background

There are ethnic differences in the care pathways and treatments people with psychosis receive. Within the UK, people of Black Caribbean and Black African descent are more likely to: enter mental health services via forensic pathways and experience compulsory detention,[1] receive medication by depot[2] and be subject to community treatment orders.[3] Black people with treatment-resistant schizophrenia are less likely to receive drug treatments in accordance with national guidelines, and Asian British people with a schizophrenia diagnosis are less likely to receive copies of their care plans.[2] Treatment inequalities based on ethnicity have also been identified in other countries. For example, in the USA, people of African descent have less money spent on their healthcare through state-funded programmes[4] and are less likely to receive medication associated with fewer side effects.[5] In the Netherlands, ethnic minority groups are more likely to be compulsorily detained for treatment and less likely to be recommended for outpatient treatment.[6]

A prospective study in the UK found significant ethnic differences in Mental Health Act 2007 (MHA) assessments and detentions, with Black Africans having higher rates than any other ethnic group.[7] However, when controlling for diagnosis, age, risk and social support, there were no significant ethnic differences in detention.[7] Similarly, Singh *et al*[8] found no significant differences between ethnic groups in MHA detention while controlling for variables such as risk and social support. These studies raise the possibility that treatment differences could be accounted for by ethnic differences in factors such as: self-harm and suicide attempt,[9] psychosis symptom profiles,[10] deprivation[11] and substance use.[12]

UK national guidelines recommend cognitive–behavioural therapy (CBT) for the treatment and prevention of psychosis (CBTp), as CBTp has demonstrated robust evidence of its efficacy on service user outcomes.[13] However, the National Audit of Schizophrenia found that CBTp was only offered to 39% of service users and accessed by 19% of service users.[14] There are evidently barriers to accessing CBTp (eg, Hazell *et al*, Prytys *et al*[15 16]), although certain factors may increase referral to CBTp (eg, higher levels of positive symptoms[17]).

People from ethnic minority communities experience additional barriers to access and engagement with psychological therapy more generally.[18] In the UK, people of Black Caribbean and Black African descent with psychosis are less likely to receive a talking therapy than their white British counterparts.[19–21] A nationally representative survey of people with psychosis found that all ethnic minority groups (excluding those with mixed ethnicity) were less likely to be offered CBT, and Black service users were less likely to be offered family therapy.[2] Similar findings have been demonstrated in international samples, where Black Americans with psychosis are less likely to receive a talking therapy than their white American counterparts.[22] Nonetheless, research emanating from the UK (South London and Maudsley Improving Access to Psychological Therapies for people with severe mental illness (SLaM IAPT-SMI) Demonstration Site) has indicated that after CBTp has been offered there is no difference between a Black and Minority ethnic (BME) group and a non-BME group in engagement in CBTp.[23 24]

Engagement is a complex concept that requires the service provider being adequately engaging and the recipient to be adequately engaged. There are potentially many explanations of ethnic variations in access to and engagement with CBT. For example, ethnic minority communities have more coercive pathways into treatment (eg, Mann *et al*[1]), which may adversely influence the therapeutic relationship,[25] and subsequently impact on engagement in treatment.[26] Other barriers to engagement might include: lower socioeconomic status[27]; increased stigma in certain communities[28]; fear of service users by providers and fear of providers by service users[29]; suspiciousness of mental health services and non-culturally appropriate therapy[30]; language barriers[31]; clinicians' perceptions of religious and spiritual explanations for psychosis[32]; and institutional racism within mental health services.[33 34]

## Research questions and rationale

There is a lack of information about the extent of inequalities experienced by ethnic minority groups with serious mental illness, despite well-recognised adverse outcomes in certain minority groups. Furthermore, there is a paucity of information about the role that risk and symptom severity plays in treatment disparity (including access to psychological therapy) for ethnic minority groups. In order to address these gaps in knowledge, using all the case records from a large secondary care mental health-care provider, this study set out to answer the following questions:

1. In people who have had a diagnosis of bipolar disorder (ICD-10 code F30-1) or psychosis (ICD-10 code F20-29 excluding F21), are there variations by ethnic group in receipt of either individual or group CBT after adjustment for differences in risk profiles and symptom severity?
2. Do ethnic group variations in receipt of CBT differ between contexts (eg, inpatient vs community settings) after adjustment for risk profiles and symptom severity?

## METHOD
### Study design and setting

The data, which were generated as part of routine care, were derived from clinical records from South London and Maudsley (SLaM) Trust. SLaM is a near-monopoly provider of secondary mental health services[35] for a catchment of over 1.2 million residents in South London and has over 400 000 service user records.[36] The SLaM catchment boroughs are not dissimilar from London as a whole in terms of age, education, gender and socioeconomic status.[36 37] However, SLaM has a higher proportion of ethnic minority groups in comparison with England as whole.[36] The (self-assigned) ethnicity population distribution recorded in the 2011 census for the SLaM catchment area is: 55.1% white, 24.7% Black, 10.8% Asian, 6.9% mixed ethnicity and 2.5% other.[36] Even after adjustment for age, sex and ethnicity, areas within SLaM's catchment have been shown to have a 2.2 times higher incidence of psychosis than the European average.[38]

This investigation used the Clinical Record Interactive Search (CRIS) tool[36] to access an anonymised data set derived from SLaM's electronic health records that comprise the Maudsley Biomedical Research Centre (BRC) Case Register. The BRC Case Register uses an opt-out mechanism, which is seldom used (circa n=4). Consequently, the sampling techniques employed ensure that persons who have not experienced good engagement with mental health services are still represented in the sample. Established in 2008, the CRIS system facilitates access and retrieval of anonymised clinical records. For a more in-depth description of how the data are stored, anonymised, and accessed see refs 36 37 39.

## Sample

Cases were included if they had received an ICD-10 diagnosis of a bipolar-related mental health problem (ie, manic episode (F30) and/or bipolar affective disorder (F31)) and were defined as having a bipolar disorder. The psychosis group included anyone with any of the following diagnoses: schizophrenia (F20), delusional disorder (F22), brief psychotic disorder (F23), shared psychotic disorder (F24), schizoaffective disorder (F25), other nonorganic psychotic disorders (F28) and unspecified nonorganic psychosis (F29).

No upper limit was set on age. Cases were excluded if: they were under the age of 15 years (a criterion that has been previously applied to this cohort[40]); they had a diagnosis of an organic/non-functional disorder; or there were missing data regarding marital status, ethnicity, Index of Multiple Deprivation (IMD) score, gender or age. To this end, only participants with complete data were included.

Due to limited numbers in some ethnic groups, cases were excluded if their recorded ethnicity did not belong to one of the following Office of National Statistics categories: Black African, Black Caribbean, Irish and white British.[41] A group labelled 'South Asian' including individuals recorded as Indian, Pakistani or Bangladeshi was also included in the sample. This investigation used the same approach of defining and grouping ethnicity that has been applied to CRIS data previously.[40 42]

## Data retrieval

SLaM adopted fully electronic health records for all its services in 2006, including the importing of legacy data. The current data set includes records from 1 January 2007 up until the extraction date of 31 July 2017. Source clinical records contain information from structured closed question response boxes (eg, age) and free text. Automated natural language processing (NLP) algorithms (see ref 43) are used to determine the presence and prescribed 'value' of variables contained in free text.

Within the current investigation, NLP algorithms were used to provide supplementary information on diagnoses and CBT. Recording an ICD-10[44] diagnosis within a structured field is mandatory within SLaM,[45] supplemented by NLP to ascertain diagnoses recorded in free-text sources, for example, clinical notes.[36 45] Another NLP algorithm has been developed to identify case notes that document a CBT session,[19] again supplementing information within structured fields and achieving in combination a positive predictive value of 95% and a sensitivity of 96%.[19]

## Demographic, clinical and treatment data extracted and operationalised

Demographic data retrieved included gender, marital status, ethnicity and age. All of the demographic data was retrieved at the point of data extraction (31 July 2017), for example, the participants' age on the 31 of July 2017. From lower super output area of residence, a standard national geographic unit containing approximately 1500 residents, area level deprivation was calculated from the IMD.[46] Multiple area level assessments contribute to seven subscales (income deprivation; employment deprivation; education, skills and training deprivation; health deprivation and disability; crime; barriers to housing and services; and living environment deprivation) that form the IMD. Scores on the IMD were split into deciles within the current sample.

The algorithm within the SLaM clinician interface ensures that structured risk assessments are completed when risk information is noted. We developed an assessment of severity and risk based on previous approaches used with this dataset.[47] To this end, we retrieved information from structured risk assessments pertaining to: history of violence, history of 'non-adherence', history of suicide attempt, perceived lethal means used in suicide attempt, current plans to end life, expression of suicidal ideation, expressed feelings of hopelessness, expressed high levels of subjective distress and expressed feelings of having no control. We also retrieved information about previous: substance use disorder diagnosis (ICD code F1), inpatient admissions, treatment under the MHA, Accident and Emergency (A&E) attendance (for mental health problems), referral to assertive outreach, referral to the crisis team and forensic history.

We retrieved data about the CBT session regarding: whether the service user was an inpatient or outpatient at the time of contact; whether the contact was face to face or remote (eg, via telephone); and whether the contact was in a one-to-one or group session. In line with national standard guidelines definition of access,[48] the current investigation assessed whether participants had at least one documented session of CBT. National Institue for Health and Clinical Excellence (NICE) guidelines for psychosis recommend that CBT is delivered 'over at least 16 planned session (sic)' (13, p. 589). NICE guidelines for bipolar disorder recommend that a depressive episode should be treated with between 16 and 20 sessions of CBT.[49] Consequently, a 16-session criterion was also adopted as a more stringent definition of a course of CBT. Jolley and colleagues[23] operationalised CBT completion as at least five sessions. Supplementary analyses were conducted using this less stringent definition of the completion of CBT treatment. Analyses of the 5 and 16 session criteria were restricted to participants who had at least one documented session of CBT (n=5197). Participants were also excluded from analyses regarding the 5 and 16 session criteria if they were currently receiving CBT at data extraction and had not received a minimum of 5 or 16 sessions of CBT, which resulted in 100 and 220 participants being excluded respectively (see figure 1). CBT that was currently ongoing was defined as anyone who had a CBT session in the 6 weeks prior to data extraction.

## Patient and public involvement

This specific project was reviewed, commented on and approved by the CRIS Oversight Committee, which is

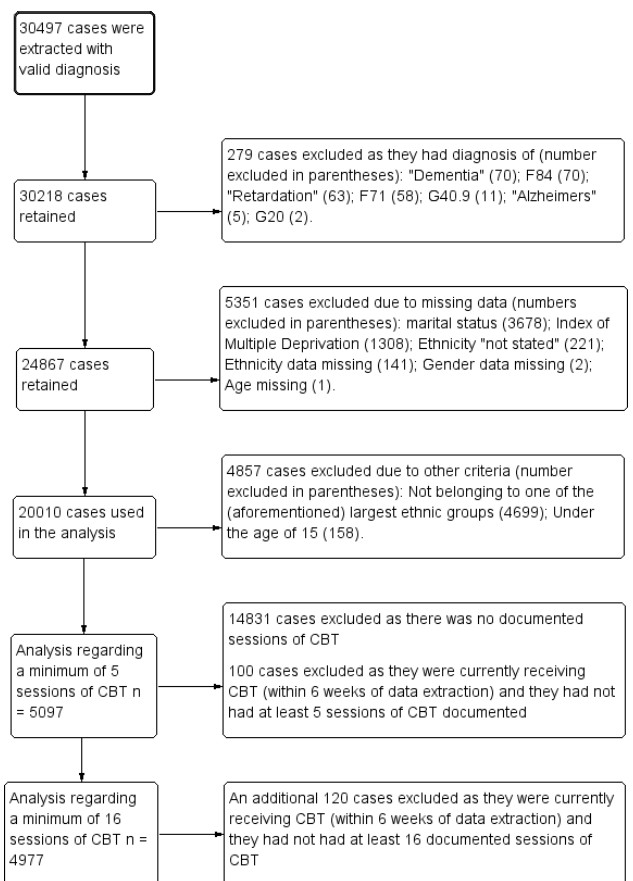

**Figure 1** Demonstrating the flow of included cases. CBT, cognitive–behavioural therapy.

chaired by a service user representative. Furthermore, the development of the CRIS system was informed by consultation with service users.[39]

### Analysis

Logistic regression models were built using multivariable procedures in Stata V.12. Models were adjusted for demographic data (gender, age, IMD and marital status), diagnoses (psychosis/bipolar disorder) and risk/severity variables (as described previously). Analyses are presented as: crude associations; adjustments for demographic data and diagnosis (step 1); and adjustments for demographic data, diagnosis and the risk/severity variables (step 2).

## RESULTS
### Descriptive statistics

A total of 5351 cases were excluded due to missing data relating to marital status (n=3678), Index of Multiple Deprivation (n=1308), ethnicity (n=362), gender (n=2) and age (n=1). The final sample consisted of 20 010 cases; figure 1 displays the flow of cases through the study.

The majority of cases were white British (n=10 393, 51.9%); the next largest ethnic group were Black Caribbean people who made up 27.4% of the sample (n=5481). There were more male cases (n=10 457, 52.3%) than

female, and the majority were single (n=17 097, 85.4%). Table 1 summarises the demographic and diagnosis data (at the time of data extraction) with relevant proportions for each ethnic group. Further information on treatment, risk and severity including items from the structured risk assessment can be found in online supplemental table 1.

Just over a quarter of the sample (26.0%, n=5197) had a documented session of CBT in the study period. The median number of sessions of CBT was 5 (IQR 13). Considering all CBT sessions documented, most were delivered face to face at a ratio of approximately 30 face-to-face sessions for every one remote (eg, telephone) session delivered and as individual rather than group sessions at a ratio of approximately 17:1. Of the people who had received CBT, 30% had their first ever (documented) session as an inpatient, 55.4% had ≥5 sessions and 25.8% had received ≥16 sessions. Further information about CBT can be found in online supplemental table 2.

### Ethnicity and reported receipt of CBT as an inpatient or outpatient

Table 2 displays the unadjusted and adjusted ORs for having a reported session of CBT in relation to ethnicity and covariates. The final adjusted model indicated that the Black African group were significantly less likely to receive CBT than the white British group (OR 0.73, 95% CI 0.66 to 0.82, p<0.001), after risk indicators were taken into account. In the adjusted model, several factors related to risk and severity were independently associated with increased likelihood of reported receipt of CBT, including lifetime inpatient admission, history of non-adherence, history of suicide attempt, lethal means used in suicide attempt, suicidal ideation, feelings of hopelessness, high levels of distress, no feelings of control and referral to the crisis team. However, a history of a substance misuse disorder diagnosis and plans to end life were associated with a decreased likelihood of reported receipt of CBT.

### Ethnicity and a minimum of 16 CBT sessions

Table 3 displays the unadjusted and adjusted ORs of receiving a minimum of 16 sessions of CBT in relation to ethnicity and covariates. The adjusted model indicated that the Black Caribbean group were significantly less likely to receive a minimum of 16 sessions of CBT than the white British group (OR 0.83, 95% CI 0.71 to 0.98, p=0.03). The model also indicated that receiving the first session of CBT as an inpatient was associated with decreased odds of having at least 16 sessions of CBT (OR 0.35, 95% CI 0.29 to 0.42, p<0.001) and some of the indicators of risk increased the odds of receiving CBT (history of suicide attempt, reported high levels of distress and lifetime referral to crisis team). However, several factors associated with increased odds of ever receiving a documented session of CBT (table 2) were not significantly associated with having a minimum of 16 documented sessions (ie, lifetime inpatient admittance, history of

**Table 1** Information on baseline demographics and diagnoses and their relevant proportions for each ethnic group

| | White British | | Irish | | Black African | | Black Caribbean | | South Asian | | Total | | χ² (df) | P value |
|---|---|---|---|---|---|---|---|---|---|---|---|---|---|---|
| | N | % | N | % | N | % | N | % | N | % | | | | |
| Ethnicity | 10393 | 52 | 570 | 3 | 2817 | 14 | 5481 | 27 | 749 | 4 | 20010 | | | |
| Female | 5070 | 49 | 269 | 47 | 1350 | 48 | 2497 | 46 | 367 | 49 | 9553 | 48 | | |
| Male | 5323 | 51 | 301 | 53 | 1467 | 52 | 2984 | 54 | 382 | 51 | 10457 | 52 | 16 | <0.01 |
| Index of Multiple Deprivation | | | | | | | | | | | | | | |
| 1 | 1489 | 14 | 42 | 7 | 70 | 3 | 199 | 4 | 59 | 8 | 1859 | 9 | 1000 (36)* | <0.001 |
| 2 | 1160 | 11 | 53 | 9 | 165 | 6 | 456 | 8 | 92 | 12 | 1926 | 10 | | |
| 3 | 1133 | 11 | 62 | 11 | 195 | 7 | 536 | 10 | 87 | 12 | 2013 | 10 | | |
| 4 | 1041 | 10 | 53 | 9 | 284 | 10 | 542 | 10 | 86 | 12 | 2006 | 10 | | |
| 5 | 980 | 9 | 58 | 10 | 302 | 11 | 584 | 11 | 82 | 11 | 2006 | 10 | | |
| 6 | 920 | 9 | 62 | 11 | 327 | 12 | 654 | 12 | 69 | 9 | 2032 | 10 | | |
| 7 | 933 | 9 | 60 | 11 | 326 | 12 | 617 | 11 | 80 | 11 | 2016 | 10 | | |
| 8 | 919 | 8.8 | 59 | 10 | 407 | 15 | 651 | 12 | 54 | 7 | 2090 | 10 | | |
| 9 | 867 | 8.3 | 60 | 11 | 379 | 14 | 646 | 12 | 64 | 9 | 2016 | 10 | | |
| 10 | 951 | 9.2 | 61 | 11 | 362 | 13 | 596 | 11 | 76 | 10 | 2046 | 10 | | |
| Single marital status | 8784 | 85 | 486 | 85 | 2300 | 82 | 5035 | 92 | 492 | 66 | 17097 | 85 | 456(4)* | <0.001 |
| In relationship | 1609 | 16 | 84 | 15 | 517 | 18 | 446 | 8 | 257 | 34 | 2913 | 15 | | |
| Age: median (IQR) | 49 (26.9) | | 56 (28.8) | | 43 (18.8) | | 46 (22.3) | | 47 (26.2) | | 48 (24.5) | | 451(4)* | <0.001 |
| Psychosis† | 6516 | 63 | 366 | 64 | 2435 | 86 | 4617 | 84 | 563 | 75 | 14497 | 73 | 1200(4)* | <0.001 |
| Bipolar‡ | 3877 | 37 | 204 | 36 | 382 | 14 | 864 | 16 | 186 | 25 | 5513 | 28 | | |
| Lifetime comorbid substance use diagnosis | 1675 | 16 | 140 | 25 | 292 | 10 | 865 | 16 | 53 | 7 | 3025 | 15.1 | 94 (4)* | <0.001 |

*Kruskal-Wallis H non-parametric test for ranked data used to determine the χ² value
†Psychosis=diagnosis of schizophrenia, delusional disorder, brief psychotic disorder, schizoaffective disorder, other non-organic psychotic disorders or unspecified non-organic psychosis.
‡Bipolar=diagnosis of a manic episode or bipolar affective disorder.
1, least deprived; 10, most deprived.

**Table 2** Crude and adjusted associations from logistic regression models for at least one recorded session of CBT (inpatient or outpatient)

| Variable | N | OR (95% CI) | | |
| --- | --- | --- | --- | --- |
| | | Crude associations | Step 1 | Step 2 |
| **Ethnicity** | | | | |
| White British | 10393 | Reference group | | |
| Irish | 570 | 1.00 (0.82 to 1.21) | 1.12 (0.91 to 1.36) | 1.05 (0.85 to 1.29) |
| Black African | 2817 | 1.06 (0.97 to 1.17) | 0.96 (0.87 to 1.06) | 0.73 (0.66 to 0.82)*** |
| Black Caribbean | 5481 | 1.29 (1.20 to 1.39)*** | 1.20 (1.11 to 1.30)*** | 0.93 (0.86 to 1.02) |
| South Asian | 749 | 0.99 (0.83 to 1.18) | 0.97 (0.82 to 1.16) | 0.93 (0.77 to 1.12) |
| **Gender** | | | | |
| Female | 9553 | Reference group | | |
| Male | 10457 | 0.89 (0.84 to 0.95)*** | 0.84 (0.78 to 0.89)*** | 0.84 (0.78 to 0.90)*** |
| Age (years) | | 0.98 (0.98 to 0.99)*** | 0.98 (0.98 to 0.99)*** | 0.99 (0.98 to 0.99)*** |
| **Area level deprivation** | | | | |
| IMD decile (per 10th) | | 1.01 (1.00 to 1.02) | 1.01 (0.99 to 1.02) | 0.99 (0.98 to 1.00) |
| **Marital status** | | | | |
| In relationship | 2913 | Reference group | | |
| Single | 17097 | 1.23 (1.12 to 1.35) | 1.08 (0.98 to 1.19) | 1.07 (0.97 to 1.18) |
| **Diagnosis** | | | | |
| Psychosis | 14497 | Reference group | | |
| Bipolar affective disorder | 5513 | 0.94 (0.88 to 1.01) | 0.93 (0.86 to 1.00) | 1.00 (0.93 to 1.09) |
| **Comorbid substance misuse** | | | | |
| No previous substance misuse diagnosis | 16985 | Reference group | | |
| Lifetime comorbid substance misuse diagnosis | 3025 | 1.31 (1.20 to 1.42)*** | | 0.85 (0.77 to 0.93)*** |
| **Admission** | | | | |
| No previous admission | 10593 | Reference group | | |
| Inpatient admission ever | 9417 | 3.20 (2.99 to 3.42)*** | | 1.76 (1.58 to 1.95)*** |
| **Treatment under the Mental Health Act (MHA)** | | | | |
| Never treated under MHA | 12904 | Reference group | | |
| Ever treated under MHA | 7106 | 2.54 (2.38 to 2.71)*** | | 0.96 (0.87 to 1.07) |
| **Structured risk assessment items†** | | | | |
| History of violence | 6216 | 2.31 (2.16 to 2.47)*** | | 1.09 (1.00 to 1.20) |
| Difficulty managing physical health | 3622 | 1.74 (1.61 to 1.88)*** | | 0.97 (0.88 to 1.07) |
| History of non-adherence | 6425 | 2.55 (2.39 to 2.73)*** | | 1.27 (1.16 to 1.39)*** |
| History of suicide attempt | 3758 | 2.83 (2.63 to 3.05)*** | | 1.36 (1.22 to 1.53)*** |
| Lethal means used in suicide attempt | 2026 | 2.65 (2.41 to 2.91)*** | | 1.04 (1.22 to 1.53)*** |
| Plans to end life | 863 | 2.62 (2.29 to 3.01)*** | | 0.82 (0.69 to 0.96)* |
| Suicidal ideation | 2041 | 3.23 (2.94 to 3.55)*** | | 1.24 (1.10 to 1.41)*** |

Continued

**Table 2** Continued

| Variable | N | OR (95% CI) Crude associations | Step 1 | Step 2 |
|---|---|---|---|---|
| Feelings of hopelessness | 2850 | 3.06 (2.82 to 3.32)*** | | 1.24 (1.11 to 1.40)*** |
| High level of distress | 4666 | 3.24 (3.02 to 3.47)*** | | 1.53 (1.40 to 1.68)*** |
| No feelings of control | 2972 | 3.03 (2.79 to 3.28)*** | | 1.22 (1.09 to 1.36)*** |
| Referred/seen by other team | | | | |
| Never referred to crisis team | 13 504 | Reference group | | |
| Ever referred to the crisis team | 6506 | 2.96 (2.77 to 3.16)*** | | 1.69 (1.57 to 1.83)*** |
| Never seen at A&E‡ | 13 389 | Reference group | | |
| Ever seen at A&E‡ | 6621 | 1.69 (1.58 to 1.80)*** | | 0.97 (0.90 to 1.04) |
| Never referred to assertive outreach | 18 977 | Reference group | | |
| Ever referred to assertive outreach | 1033 | 1.51 (1.32 to 1.72)*** | | 0.94 (0.81 to 1.09) |
| Forensic history | | | | |
| No forensic history reported | 18 137 | Reference group | | |
| Forensic history reported | 1873 | 1.70 (1.53 to 1.88)*** | | 1.07 (0.96 to 1.20) |

Step 1: adjusted for ethnicity+gender+age+IMD decile+marital status+diagnosis: psychosis/bipolar.
Step 2: adjusted for ethnicity+gender+age+IMD decile+marital status+diagnosis: psychosis/bipolar+substance use diagnosis+inpatient admittance+treated under the MHA+structured risk assessment items (entered separately)+referred to crisis team+treated at A&E+referred to assertive outreach+forensic history.
*P<0.05;**p<0.01; ***p<0.001.
†For brevity, reference groups are omitted. Reference groups are a non-affirmative response to the item. The n for the reference group is the number of people included in the analysis (n=20 010) – the number of people with an affirmative response.
‡Seen at A&E due to mental health emergency.
1, least deprived; 10, most deprived; CBT, cognitive–behavioural therapy; IMD, Index of Multiple Deprivation.

non-adherence, lethal means used in suicide attempt, reported suicidal ideation, reported feelings of hopelessness and reported feelings of a lack of control).

### Ethnicity and reported receipt of CBT as an inpatient

Analyses were restricted to participants who had been an inpatient (n=9417) and associations investigated with receipt or not of CBT in this setting. Unadjusted and adjusted associations are displayed in table 4. The adjusted model demonstrated that the Black African group (OR 0.76, 95% CI 0.65 to 0.89, p=0.001) and the Black Caribbean group (OR 0.83, 95% CI 0.73 to 0.94, p=0.003) were significantly less likely to have received CBT than the white British group.

### Supplementary analyses

Analyses using the less stringent definition of a course of CBT (≥5 sessions) indicated the Black African group were significantly less likely to receive this in comparison to the white British group (OR 0.76, 95% CI 0.63 to 0.91, p=0.003) (see online supplemental table 3). Analyses of CBT sessions received only as an outpatient also indicated that the Black African group (OR 0.75, 95% CI 0.67 to 0.84, p<0.001) were significantly less likely to receive this

than the white British group (see online supplemental table 4).

### Post hoc sensitivity analysis
### Recording of clinical risk

The crude estimates indicated that several variables indicative of higher clinical risk and severity were associated with increased odds of having a (single) documented session of CBT (table 2). We considered that this may be because CBT is better recorded (rather than more likely to be delivered) for those at an increased risk (eg, of harming themselves, suicide and harming others) and proposed that, if defensive practice resulted in better note keeping, this would be most likely evident in the structured fields. Consequently, as a supplementary sensitivity analysis, using the entire sample (n=20 010), models assessing reported receipt of CBT were rerun omitting entries identified in the structured fields, (ie, just using data derived from free text). However, this analysis continued to indicate an association between Black African group membership and significantly lower odds of receiving CBT than white British group membership (OR 0.76, 95% CI 0.63 to 0.92, p=0.004). Adjusted and

**Table 3** Crude and adjusted associations from logistic regression models for at least 16 recorded sessions of CBT

| Variable | N | OR (95% CI) | | |
| --- | --- | --- | --- | --- |
| | | Crude associations | Step 1 | Step 2 |
| **Ethnicity** | | | | |
| White British | 2456 | Reference group | | |
| Irish | 137 | 1.03 (0.70 to 1.50) | 1.02 (0.70 to 1.50) | 1.05 (0.71 to 1.55) |
| Black African | 682 | 0.78 (0.64 to 0.95)* | 0.77 (0.63 to 0.95)* | 0.86 (0.69 to 1.06) |
| Black Caribbean | 1524 | 0.77 (0.67 to 0.90)** | 0.76 (0.65 to 0.89)** | 0.83 (0.71 to 0.98)* |
| South Asian | 178 | 0.98 (0.70 to 1.38) | 0.99 (0.72 to 1.39) | 1.03 (0.73 to 1.47) |
| **Gender** | | | | |
| Female | 2485 | Reference group | | |
| Male | 2492 | 0.99 (0.87 to 1.12) | 0.98 (0.86 to 1.11) | 1.05 (0.91 to 1.20) |
| **Age (years)** | | 1.00 (1.00 to 1.01) | 1.00 (1.00 to 1.01) | 1.00 (1.00 to 1.01) |
| **Area level deprivation** | | | | |
| IMD decile (per 10th) | | 0.99 (0.97 to 1.01) | 1.00 (0.97 1.02) | 0.99 (0.97 to 1.01) |
| **Marital status** | | | | |
| In relationship | 639 | Reference group | | |
| Single | 4338 | 1.07 (0.88 to 1.29) | 1.11 (0.91 to 1.36) | 1.21 (0.98 to 1.48) |
| **Diagnosis** | | | | |
| Psychosis | 3645 | Reference group | | |
| Bipolar affective disorder | 1332 | 0.95 (0.83 to 1.10) | 0.90 (0.77 to 1.04) | 0.86 (0.74 to 1.01) |
| **Comorbid substance misuse** | | | | |
| No previous substance misuse diagnosis | 4090 | Reference group | | |
| Lifetime comorbid substance misuse diagnosis | 887 | 0.81 (0.69 to 0.97)* | | 0.79 (0.66 to 0.96)* |
| **Admission** | | | | |
| No previous admission | 1622 | Reference group | | |
| Inpatient admission ever | 3355 | 0.74 (0.65 to 0.85)*** | | 1.06 (0.86 to 1.31) |
| **Treatment under Mental Health Act (MHA)** | | | | |
| Never treated under MHA | 2429 | Reference group | | |
| Ever treated under the MHA | 2548 | 0.70 (0.61 to 0.79)*** | | 0.86 (0.71 to 1.05) |
| **Structured risk assessment items†** | | | | |
| History of violence | 2234 | 0.80 (0.71 to 0.91)** | | 0.93 (0.78 to 1.10) |
| Difficulty managing physical health | 1237 | 0.94 (0.81 to 1.09) | | 1.01 (0.85 to 1.20) |
| History of non-adherence | 2382 | 0.83 (0.73 to 0.95)** | | 0.91 (0.77 to 1.08) |
| History of suicide attempt | 1589 | 1.39 (1.22 to 1.59)*** | | 1.33 (1.09 to 1.61)** |

**Table 3** Continued

| Variable | N | OR (95% CI) | | |
| --- | --- | --- | --- | --- |
| | | Crude associations | Step 1 | Step 2 |
| Lethal means used in suicide attempt | 887 | 1.36 (1.16 to 1.60)*** | | 1.01 (0.80 to 1.27) |
| Reported plans to end life | 382 | 1.54 (1.23 to 1.92)*** | | 1.33 (1.01 to 1.73)* |
| Suicidal ideation | 961 | 1.38 (1.18 to 1.61)*** | | 1.10 (0.89 to 1.35) |
| Feelings of hopelessness | 1287 | 1.32 (1.14 to 1.52)*** | | 1.01 (0.82 to 1.23) |
| High level of distress | 2000 | 1.22 (1.07 to 1.39)** | | 1.22 (1.03 to 1.44)* |
| No feelings of control | 1337 | 1.24 (1.08 to 1.43)** | | 1.09 (0.90 to 1.31) |
| Referred/seen by other team | | | | |
| Never referred to crisis team | 2459 | Reference group | | |
| Ever referred to the crisis team | 2518 | 1.27 (1.12 to 1.44)*** | | 1.34 (1.14 to 1.56)*** |
| Never seen at A&E‡ | 2918 | Reference group | | |
| Ever seen at A&E‡ | 2059 | 0.96 (0.84 to 1.09) | | 0.93 (0.80 to 1.08) |
| Never referred to assertive outreach | 4636 | Reference group | | |
| Ever referred to assertive outreach | 341 | 0.67 (0.51 to 0.89)** | | 0.81 (0.60 to 1.08) |
| Forensic history | | | | |
| No forensic history reported | 4326 | Reference group | | |
| Forensic history reported | 651 | 0.80 (0.66 to 0.98)** | | 0.86 (0.69 to 1.06) |
| Context of first CBT session | | | | |
| First CBT as outpatient | 3493 | Reference group | | |
| First CBT as inpatient | 1484 | 0.35 (0.29 to 0.41) *** | | 0.35 (0.29 to 0.42) *** |

Step 1: adjusted for ethnicity+gender+age+IMD decile+marital status+diagnosis: psychosis/bipolar.

Step 2: adjusted for ethnicity+gender+age+IMD decile+marital status+diagnosis: psychosis/bipolar+substance use diagnosis+inpatient admittance+treated under the MHA+structured risk assessment items (entered separately)+referred to crisis team+treated at A&E+referred to assertive outreach+forensic history+first CBT as inpatient.

*P<0.05; **p<0.01; ***p<0.001.

†For brevity, reference groups are omitted. Reference groups are a non-affirmative response to the item. The n for the reference group is the number of people included in the analysis (N=4977) – the number of people with an affirmative response.

‡Seen at A&E due to mental health emergency.

1, least deprived; 10, most deprived; CBT, cognitive–behavioural therapy; IMD, Index of Multiple Deprivation.

unadjusted ORs are presented in online supplemental table 5.

### Influence of time

Additional analyses were conducted to assess if changes over time affected referral practices for psychological treatments. To this end, a variable was created indicating participants who had received a diagnosis of psychosis or bipolar affective disorder after the midpoint of the data collection window (ie, after 16 April 2012). Models considering ethnicity and reported receipt of CBT were rerun including the variable indicating the date at which diagnosis was given. This analysis also indicated that the Black African group were significantly less likely to receive CBT than the white British group (OR 0.72, 95% CI 0.65 to 0.81, p<0.001), suggesting that this finding was not influenced by the date diagnosis was given (see online supplemental table 6). In the fully adjusted model, receiving a diagnosis of psychosis or bipolar affective disorder after the midpoint of the data collection window was associated with decreased odds of a documented session of CBT

**Table 4** Crude and adjusted associations from logistic regression models for at least one recorded session of CBT as an inpatient

| Variable | N | OR (95% CI) | | |
| --- | --- | --- | --- | --- |
| | | Crude associations | Step 1 | Step 2 |
| **Ethnicity** | | | | |
| White British | 4000 | Reference group | | |
| Irish | 232 | 0.95 (0.69 to 1.32) | 1.02 (0.73 to 1.41) | 0.99 (0.71 to 1.39) |
| Black African | 1734 | 0.82 (0.71 to 0.95)** | 0.80 (0.69 to 0.93)** | 0.76 (0.65 to 0.89)** |
| Black Caribbean | 3132 | 0.93 (0.83 to 1.05) | 0.91 (0.80 to 1.02) | 0.83 (0.73 to 0.94)** |
| South Asian | 319 | 0.82 (0.62 to 1.10) | 0.83 (0.62 to 1.12) | 0.86 (0.64 to 1.16) |
| **Gender** | | | | |
| Female | 4390 | Reference group | | |
| Male | 5027 | 0.93 (0.84 to 1.03) | 0.89 (0.80 to 0.99) | 0.87 (0.79 to 0.97) |
| Age (years) | | 0.99 (0.99 to 1.00)*** | 0.99 (0.99 to 1.00)*** | 0.99 (0.99 to 0.99)*** |
| **Area-level deprivation** | | | | |
| IMD decile (per 10th) | | 0.97 (0.95 to 0.99)** | 0.97 (0.96 to 0.99)** | 0.97 (0.95 to 0.99)** |
| **Marital status** | | | | |
| In relationship | 1234 | Reference group | | |
| Single | 8183 | 1.24 (1.06 to 1.45)** | 1.19 (1.02 to 1.40)* | 1.08 (0.91 to 1.27) |
| **Diagnosis** | | | | |
| Psychosis | 7114 | Reference group | | |
| Bipolar affective disorder | 2303 | 0.97 (0.86 to 1.09) | 0.94 (0.83 to 1.06) | 1.02 (0.90 to 1.16) |
| **Comorbid substance misuse** | | | | |
| No previous substance misuse diagnosis | 7456 | Reference group | | |
| Lifetime comorbid substance misuse diagnosis | 1961 | 1.05 (0.93 to 1.19) | | 0.88 (0.77 to 1.00) |
| **Treatment under Mental Health Act (MHA)** | | | | |
| No treatment under MHA | 2506 | Reference group | | |
| Ever treated under MHA | 6911 | 1.56 (1.38 to 1.76)*** | | 1.39 (1.21 to 1.59)*** |
| **Structured risk assessment items** | | | | |
| History of violence | 4914 | 1.56 (1.41 to 1.73)*** | | 1.13 (1.00 to 1.28)* |
| Difficulty managing physical health | 2720 | 1.59 (1.44 to 1.77)*** | | 1.34 (1.19 to 1.51)*** |
| History of non-adherence | 5161 | 1.66 (1.50 to 1.84)*** | | 1.24 (1.09 to 1.41)** |
| History of suicide attempt | 2879 | 1.61 (1.46 to 1.79)*** | | 1.17 (1.00 to 1.35)* |
| Lethal means used in suicide attempt | 1612 | 1.56 (1.38 to 1.77)*** | | 1.02 (0.86 to 1.20) |
| Plans to end life | 754 | 1.66 (1.41 to 1.96)*** | | 1.09 (0.89 to 1.32) |
| Suicidal ideation | 1684 | 1.66 (1.47 to 1.87)*** | | 1.14 (0.97 to 1.33) |
| Feelings of hopelessness | 2218 | 1.66 (1.48 to 1.85)*** | | 1.08 (0.93 to 1.25) |
| High level of distress | 3747 | 1.82 (1.65 to 2.02)*** | | 1.37 (1.22 to 1.54)*** |
| No feelings of control | 2370 | 1.68 (1.51 to 1.87) * | | 1.08 (0.94 to 1.24) |
| **Referred/seen by other team** | | | | |
| Never referred to crisis team | 4217 | Reference group | | |
| Ever referred to the crisis team | 5200 | 1.08 (0.97 to 1.19) | | 0.90 (0.80 to 1.00)* |
| Never seen at A&E | 4981 | Reference group | | |

Continued

**Table 4** Continued

| Variable | N | OR (95% CI) | | |
| --- | --- | --- | --- | --- |
| | | Crude associations | Step 1 | Step 2 |
| Ever seen at A&E | 4436 | 1.22 (1.10 to 1.34) *** | | 1.11 (1.00 to 1.23) |
| Never referred to assertive outreach | 8633 | Reference group | | |
| Ever referred to assertive outreach | 784 | 1.45 (1.23 to 1.71) *** | | 1.18 (0.99 to 1.41) |
| Forensic history | | | | |
| No forensic history reported | 7936 | Reference group | | |
| Forensic history reported | 1481 | 1.11 (0.97 to 1.27) | | 1.02 (0.89 to 1.18) |

Step 1: adjusted for ethnicity+gender+age+IMD decile+marital status+diagnosis: psychosis/bipolar.
Step 2: adjusted for ethnicity+gender+age+IMD decile+marital status+diagnosis: psychosis/bipolar+substance use diagnosis+treated under the MHA+structured risk assessment items (entered separately)+referred to crisis team+treated at A&E+referred to assertive outreach+forensic history.
*P<0.05; **p<0.01; ***p<0.001.
†For brevity, reference groups are omitted. Reference groups are a non-affirmative response to the item. The n for the reference group is the number of people included in the analysis (N=9417) – the number of people with an affirmative response.
‡Seen at A&E due to mental health emergency.
1, least deprived; 10, most deprived; CBT, cognitive–behavioural therapy; IMD, Index of Multiple Deprivation.

(OR 0.77, 95% CI 0.71 to 0.83, p<0.001). Furthermore, analysis was conducted to assess if there was an interaction between time and ethnicity; however, a likelihood ratio test indicated that fitting this interaction term did not significantly improve the model: $\chi^2$ (4)=5.25, p=0.26.

## DISCUSSION
### Statement of principal findings
This investigation found that after adjustment for numerous indicators of risk and severity, in comparison with white British counterparts, Black African people with bipolar disorder or psychosis were less likely to have a documented session of CBT, a finding that was robust to a number of sensitivity analyses. After adjustment for indicators of risk and symptom severity in comparison with white British people, Black Caribbean people were also less likely to receive CBT as inpatients and were less likely to receive the minimum 16 sessions recommended by national guidelines. This study also found that regardless of ethnicity, people who had their first documented session of CBT as an inpatient were less likely to receive a minimum of 16 sessions of CBT (and a similar effect was also noted in supplementary analyses of a minimum five documented sessions and documented receipt of CBT as an outpatient). In addition, regardless of ethnicity indicators of higher risk and severity of symptoms were typically associated with higher odds of receiving CBT; however, these associations between risk status and receipt of CBT were less consistent in analyses of a minimum 16 documented sessions.

### Strengths and limitations of the study
To our knowledge, this study has used the largest sample to date to assess ethnic differences in access to CBT for people with psychosis or bipolar affective disorder. This study used a case register from a large mental healthcare provider serving a socially and ethnically diverse geographic catchment. Furthermore, the data were sourced from the full electronic health record, using a case register with near-complete coverage of people receiving mental healthcare for these diagnoses. The study used a tool to extract information about CBT from structured fields and free text, an approach that has been shown to have high positive predictive value and sensitivity values in previous work.[19] Consequently, this study likely provides a highly accurate picture of access to CBT delivered by mental health services within the catchment. Of note, despite having recognised high incidence rates of psychosis,[38] the catchment is not dissimilar to other parts of London and UK urban areas on several sociodemographic metrics[36 37]; the results of this investigation may generalise to other urban and semiurban multicultural areas in England, a notion that is supported by ethnic disparity in access to therapy indicated in nationally representative data.[2] By accessing a large data set of complete clinical records, we were able to contribute novel findings relating to the impact of risk and pathways on engagement with CBT. However, one limitation of this investigation is that it was not possible to extract information from the BRC Case Register about other psychological therapies, some of which are recommended by national guidelines and delivered routinely within the services analysed (eg, family intervention[13]). It is possible therefore that disparity in access to CBT may be accounted for by ethnic differences in preference for therapy type, although this has not been suggested to be the case in other studies of national data from the UK.[2] Another limitation is that although this study likely displays an accurate picture of service users who *received* CBT, it was not possible to derive information about the *offer* of CBT. If service users are not accepting CBT or completing a course, or alternatively service providers are not offering or delivering a course

of CBT, it is important to understand why. This could be explored in future research.

An additional limitation of this study is we did not extract information regarding the length of inpatient stay. The consequence of this is we do not know the impact of length of stay on the likelihood that someone receives CBT. It is feasible that people who have very short inpatient stays are less likely to receive CBT than those who spend longer in that environment.

### Strengths of this study in relation to other research

Our findings replicate those observed for unselected community residents from a nationally representative sample, namely less equitable access to CBT for ethnic minority groups.[2] Previous investigations that have explored ethnic disparities in access/engagement with CBT in samples with psychosis have not differentiated between Black African and Black Caribbean people,[2 19 23 24] despite the two groups typically having different migratory histories and different factors influencing pathways into treatment for psychosis.[50] The current investigation was able to define more specific ethnic categories providing a more nuanced understanding of ethnicity and access to CBT.

### Comparisons with previous research

Previous research has highlighted that more positive symptoms in psychosis increase referrals for CBT.[17] Our study extended this finding by highlighting that numerous indicators of higher symptom severity and risk increase the propensity to receive a minimum of one session of CBT. However, despite controlling for these variables, this study found persistent disparities by ethnicity in receipt of CBT (ie, a minimum of one documented session). The relationship between risk and CBT engagement (ie, documented receipt of a minimum of 16 sessions) appeared less consistent. Several of the risk indicators that increased the odds of receiving one documented session of CBT were not significantly associated either way with receipt of a minimum of 16 sessions. This may suggest a more complex relationship between risk and CBT engagement. The positive association between recorded level of clinical risk and receipt of CBT is in contrast to research suggesting that inequalities between ethnic groups in mental health treatment could be caused by differences in symptom severity.[7 8] Despite risk indicators (typically) increasing access to CBT and previous investigations suggesting that Black women are most likely to self-harm[51]; the current investigation does not indicate that ethnic disparities in the receipt of CBT is as a consequence of ethnic differences in risk or symptom profile.

First, access of CBT as an inpatient was associated with lower odds of receiving further CBT sessions. There are numerous potential explanations. For example, coercive practice in inpatient settings has been well documented, and this may potentially impact on subsequent engagement.[52] Alternatively, our finding may be related to differences in recovery styles.[53] An avoidant recovery style (referred to as sealing over) has been linked to poorer engagement with services,[54] and it is possible that some people are receptive to psychological therapy at the point of crisis (ie, during inpatient stay), but once there is a diminution of symptoms, they 'seal over' that reduces engagement.

### Implications of this research and suggestions for future research

Our study suggests that, within clinical settings, further work is needed to ensure there is parity in access to CBT. In practice, this might include ensuring that CBT is systematically offered to groups who are less likely to receive treatment. It is also feasible that further work is needed to ensure that CBT is more acceptable to Black groups that might be achieved by culturally adapting interventions.[55] Nonetheless, more research is required to explore the reasons underpinning ethnicity difference in access to CBT, whether ethnic differences in receipt of CBT extend to the offer of CBT, and the impact clinical risk has on engagement with CBT. Moreover, further research is necessary to explore the impact of pathways into care or psychological treatment and its role in subsequent engagement.

**Author affiliations**
[1]Division of Health Research, Lancaster University, Lancaster, UK
[2]Lancashire Care NHS Foundation Trust, Preston, UK
[3]Pennine Care NHS Foundation Trust, Greater Manchester, England
[4]Division of Psychology & Mental Health, School of Health Sciences, The University of Manchester, Manchester, UK
[5]Institute of Psychiatry, Psychology and Neuroscience, King's College London, London, UK
[6]South London and Maudsley NHS Foundation Trust, London, UK
[7]Section of Epidemiology, Department of Health Service & Population Research, King's College London, Institute of Psychiatry, London, UK

**Acknowledgements** We are grateful to Hitesh Shetty for his work extracting data from the Biomedical Research Centre (BRC) case register and his advice on usage of BRC Clinical Record Interactive Search (CRIS) data. We are grateful to Megan Pritchard for her advice on usage of BRC CRIS data and her support in training RMM on accessing CRIS data. We are grateful to Dr Emmanuelle Peters for her advice about the study and her advice about psychological therapy service delivery within South London and Maudsley.

**Contributors** RMM, JD-M, WS and DE made substantial contributions to the conception and design of the work and interpretation of the data. RMM and JD-M made a substantial contribution to the analysis of the data. RS made a substantial contribution to the conception and design of the work and acquisition of the data. CCo made a substantial contribution to the acquisition of the data. CCu made a substantial contribution to the interpretation of the data. All authors contributed to reviewing, drafting and revising of the manuscript. All authors have provided their approval for the work to be published and are in agreement to be accountable for all aspects of the work in ensuring that questions related to the accuracy or integrity of any part of the work are appropriately investigated and resolved.

**Funding** JD is funded by the Health Foundation working together with the Academy of Medical Sciences, for a Clinician Scientist Fellowship and by the ESRC in relation to the SEP-MD study [ES/S002715/1] and part supported by the ESRC Centre for Society and Mental Health at King's College London [ESRC Reference: ES/S012567/1]. RS is part-funded by the National Institute for Health Research (NIHR) Biomedical Research Centre at South London and Maudsley NHS Foundation Trust and King's College London, and by an NIHR Senior Investigator award. RM is funded by Lancashire Care Foundation Trust. The authors declare that the

study funders have had no role in the study design; in the collection, analysis and interpretation of the data; in the writing of the report; and in the decision to submit the paper for publication. This paper represents independent research funded by the National Institute for Health Research (NIHR) Biomedical Research Centre at South London and Maudsley NHS Foundation Trust and King's College London. The views expressed are those of the author(s) and not necessarily those of the NHS, the NIHR or the Department of Health and Social Care.

**Competing interests** SLaM and its services have had no role, in the study design, in the analysis and interpretation of the data, in the writing of the report, or in the decision to submit the paper for publication. Caroline Cupitt is employed by SLaM and works within one of SLaM's services which has produced some of the clinical notes that form part of the data analysed herein. RS declares research funding within the last 5 years from Roche, GSK and Janssen.

**Patient consent for publication** Not required.

**Ethics approval** The anonymised dataset has been approved by the NHS REC for secondary analysis (Oxford C Research Ethics Committee, reference18/SC/0372). This particular project received ethical approval from the Lancaster University Faculty of Health and Medicine Research Ethics Committee and approval from the CRIS Oversight Committee.

**Provenance and peer review** Not commissioned; externally peer reviewed.

**Data availability statement** Data may be obtained from a third party and are not publicly available. Data are owned by the South London and Maudsley NHS Foundation Trust (SLaM) which provides access to anonymised data derived from electronic medical records via the Clinical Record Interactive Search (CRIS) system. These data can only be accessed by permitted individuals from within a secure firewall (i.e. remote access is not possible and the data cannot be sent elsewhere) in the same manner as the authors. For data requests please contact Cris. administrator@slam.nhs.uk.

**ORCID iD**
Rohan Michael Morris http://orcid.org/0000-0003-1588-9845

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
