## [Reviewer comments · BMJ Open]

ARTICLE DETAILS

TITLE (PROVISIONAL)	Ethnicity and impact on the receipt of Cognitive Behavioural Therapy in people with psychosis or bipolar disorder: An English cohort study
AUTHORS	Morris, Rohan; Sellwood, William; Edge, Dawn; Colling, Craig; Stewart, Robert; Cupitt, Caroline; Das-Munshi, Jayati

VERSION 1 - REVIEW

REVIEWER	farooq naeem The University of Toronto, Toronto, Canada
REVIEW RETURNED	10-Nov-2019

GENERAL COMMENTS	This is a very well written paper that addresses an important issue. The only thing I suggest is to add details of the settings to give the readers an idea of ethnic population distribution in South London.
--

REVIEWER	Louise Johns Oxford Health NHS Foundation Trust and University of Oxford, UK.
REVIEW RETURNED	25-Nov-2019

GENERAL COMMENTS	This is an interesting and thought-provoking paper using anonymised patient record data to examine the relationship between ethnicity and receipt of CBT in service users with affective and non-affective psychosis. I have some specific comments and queries. Title: What is the meaning of implementation – is this offer of therapy, or receipt, or both? The authors clarify this later – it is receipt. Therefore, I suggest changing the title to "...receipt of CBT..." to reflect the data. Abstract: Objective – again, accessing CBT - is this being referred for or receiving CBT? Only the latter can be measured from the records (and the authors clarify later), so I suggest changing "accessing" to "receiving", as access (availability of therapy) may be the same. The conclusion regarding context (inpatient versus community settings) is unclear. Does context of therapy impact on the disparity or explain it, or is the disparity evident in both contexts? This is elaborated on further in the paper, but it would be good to clarify in the abstract.
--

	Introduction: I think it is worth referencing the SLaM IAPT-SMI Psychosis Demonstration Site (Jolley et al, 2015; Johns, Jolley et al, 2019), which found no significant demographic inequity in therapy engagement or outcomes (although ethnicity was dichotomised into BME and non-BME). For research question 1, I would specify whether this is receipt of individual CBT or individual/group CBT. Group CBT was evaluated in the IAPT-SMI Bipolar Demonstration Site (Jones et al, 2018). Method: What was the rationale for including cases from age 15? Was there an upper age limit? P11: the Jolley et al (2015) paper operationalises a “dose” of CBT (not a course of CBT) as at least 5 sessions. Results: Supplementary data 2, table 2: looking at the data, it is unclear where the differences lie. For example, CBT on-going is higher in the African and Caribbean groups and lower in the South Asian group. Similarly, there don't seem to be differences in favour of the White British group for CBT inpatient ever or CBT outpatient ever. There looks to be a difference in number of sessions, with fewer service users in the African and Caribbean groups attending more than 5 sessions. These results are clearer in tables 2 and 3, with the unadjusted ORs showing that the Caribbean group has increased odds of receiving CBT, but not in the final adjusted model. I wonder whether a different analysis strategy, e.g. all significant IVs in the unadjusted analysis entered together to examine the independent predictors of CBT receipt/sessions, would give the same results. The results regarding number of sessions received seem to be more robust than ever receiving CBT, suggesting something about engagement with CBT over time (or the mode of CBT delivery, see below). Why was length of inpatient admission not included in the analyses? This variable may impact on whether CBT was received as an inpatient. This is easy to extract from the clinical notes using admission and discharge dates. Were the authors able to check whether the CBT received as an inpatient was individual or group, as groups tend to be offered on the ward and involve fewer sessions. Discussion: As mentioned, a key limitation is that the study was not able to extract information about the offer of CBT. However, if service users are not accepting CBT, or are dropping out after fewer sessions, then it is important to understand why.
--	--

REVIEWER	Edimansyah Abdin Institute of Mental Health, Singapore
REVIEW RETURNED	18-Jan-2020

GENERAL COMMENTS	This study aims to explore the role of ethnicity in accessing Cognitive Behavioural Therapy (CBT) for people with psychosis or bipolar disorder whilst adjusting for differences in risk profiles and symptom severity and assess whether the context of treatment (inpatient versus community) impacts on the relationship between
---

	ethnicity and access to CBT using case-register data from one catchment area in London. The study has important implication for mental health services in London and elsewhere. I have few comments on how the data were analysed. My comments are as follows.  1. In order to adjust for differences in risk profiles and symptom severity effects in the analysis, i was wondering if authors can consider to include interaction effects between ethnicity and other covariates that are significantly related to the outcomes derived from Step 2 analysis. This analysis to ensure the certain group i.a younger age group, single, male, substance abuser etc within each ethnicity group may have different impact on CBT outcomes? Have authors perform multiple comparisons test on ethnicity group to examine whether other pair group comparison i.e Irish vs African, Irish vs the Caribbean, Irish vs South Asian, African vs Caribbean, African vs South Asian, Asian vs the Caribbean etc are significant? and whether the significant p values are corrected to control for type 1 error due to multiple comparisons? 2. I am not sure why Area Level Deprivation (IMD) was treated as continuous variable in regression analysis instead of as a categorical variable. Can authors clarify why they decided to decile split into decile instead of other method and keep the original categories in the regression model? 3. It is not clear about missing data and how the data is handled in the analysis. It would be good if authors can elaborate more in method section.
--	---

VERSION 1 – AUTHOR RESPONSE

Reviewer: 1

Comment:

This is a very well written paper that addresses an important issue. The only thing I suggest is to add details of the settings to give the readers an idea of ethnic population distribution in South London.

Response:

We have added the following information into the ‘Study Design and Setting’ subsection: “The (self-assigned) ethnicity population distribution recorded in the 2011 census for the SLAM catchment area is: 55.1% White, 24.7% Black, 10.8% Asian, 6.9% Mixed ethnicity, and 2.5% Other.” (p.8)

Reviewer: 2

Title:

Comment:

What is the meaning of implementation – is this offer of therapy, or receipt, or both? The authors clarify this later – it is receipt. Therefore, I suggest changing the title to “...receipt of CBT...” to reflect the data.

Response:

We have updated the title accordingly. The title now reads: “Ethnicity and impact on the receipt of Cognitive Behavioural Therapy in people with psychosis or bipolar disorder: An English cohort study”

Abstract:

Comment:

Objective – again, accessing CBT - is this being referred for or receiving CBT? Only the latter can be measured from the records (and the authors clarify later), so I suggest changing “accessing” to “receiving”, as access (availability of therapy) may be the same.

Response:

We have updated the abstract accordingly the objective now reads: “Objectives: 1) To explore the role of ethnicity in receiving Cognitive Behavioural Therapy” (p.3)

Comment:

The conclusion regarding context (inpatient versus community settings) is unclear. Does context of therapy impact on the disparity or explain it, or is the disparity evident in both contexts? This is elaborated on further in the paper, but it would be good to clarify in the abstract.

Response:

The abstract has been changed so as to not infer a direction of effect. The abstract now reads: “This study also highlights that in this cohort the context of therapy (inpatient versus community settings) has a relationship with disparity in access to treatment.” (p.4)

Introduction:

Comment:

I think it is worth referencing the SLAM IAPT-SMI Psychosis Demonstration Site (Jolley et al, 2015; Johns, Jolley et al, 2019), which found no significant demographic inequity in therapy engagement or outcomes (although ethnicity was dichotomised into BME and non-BME).

Response:

We have cited these studies in the introduction: “Nonetheless, research emanating from the UK (SLAM IAPT-SMI Demonstration Site) has indicated that after CBTp has been offered there is no difference between a Black and Minority Ethnic (BME) group and a non-BME group in engagement in CBTp.[23, 24]” (p.6)

Comment:

For research question 1, I would specify whether this is receipt of individual CBT or individual/group CBT. Group CBT was evaluated in the IAPT-SMI Bipolar Demonstration Site (Jones et al, 2018).

Response:

The research question has been updated to clarify this point. “In people who have had a diagnosis of bipolar disorder (ICD-10 code F30-1) or psychosis (ICD-10 code F20-29 excluding F21), are there variations by ethnic-group in receipt of either individual or group CBT after adjustment for differences in risk profiles and symptom severity?” (p.7)

Method:

Comment:

What was the rationale for including cases from age 15? Was there an upper age limit?

Response:

Further information is provided in text: “No upper limit was set on age. Cases were excluded if: they were under the age of 15 (a criterion which has been previously applied to this cohort;[39])” (p. 8)

Comment:

P11: the Jolley et al (2015) paper operationalises a “dose” of CBT (not a course of CBT) as at least 5 sessions.

Response:

Thank you for this comment, our understanding of this paper is that the investigators referred to therapy “completion” / “completed” or “dropout” according to that criterion. We have updated our paper to reflect the wording used in the Jolley et al., (2015) paper. Please see page 11.

Results:

Comment:

Supplementary data 2, table 2: looking at the data, it is unclear where the differences lie. For example, CBT on-going is higher in the African and Caribbean groups and lower in the South Asian group. Similarly, there don't seem to be differences in favour of the White British group for CBT

inpatient ever or CBT outpatient ever. There looks to be a difference in number of sessions, with fewer service users in the African and Caribbean groups attending more than 5 sessions.

Response:

The tables in the supplementary material have been provided to allow interested readers the possibility of assessing crude proportions of people receiving CBT by ethnicity. These are unadjusted proportions so do not take into account confounding factors (e.g. age, sex and other variables, such as previous admissions and comorbid substance use disorders) which the substantive models in the paper display. The adjusted models in the paper suggest clear trends by ethnicity in the receipt of psychological treatments.

Comment:

These results are clearer in tables 2 and 3, with the unadjusted ORs showing that the Caribbean group has increased odds of receiving CBT, but not in the final adjusted model. I wonder whether a different analysis strategy, e.g. all significant IVs in the unadjusted analysis entered together to examine the independent predictors of CBT receipt/sessions, would give the same results. The results regarding number of sessions received seem to be more robust than ever receiving CBT, suggesting something about engagement with CBT over time (or the mode of CBT delivery, see below).

Response:

We used a modelling strategy which adjusted for a priori social/ demographic indicators, followed by adjustment for clinical risk indicators in the second step. The approach is based on findings from our own and other previous work which have indicated that all of the variables which we included in the models are important to adjust for, when attempting to gain an understanding of the associations. We have retained variables in final models despite the findings in bivariable analysis (i.e. irrespective of whether the association was 'statistically significant') as we believe selecting variables on the basis of an arbitrary p value cut off leads to excluding potentially important confounders from final models, leading to systematically biased findings. See these articles which discuss the problem in taking this approach to variable selection ([https://www.jclinepi.com/article/0895-4356\(96\)00025-X/pdf](https://www.jclinepi.com/article/0895-4356(96)00025-X/pdf) Sun et al J Clin Epidemiol Vol 49 No 8 pp907-916; 1996) also <https://onlinelibrary.wiley.com/doi/pdf/10.1111/tri.12895> Heinz G. & Dinkler D. Five myths about variable selection. Transplant international 30: p6-10; 2017.

Comment:

Why was length of inpatient admission not included in the analyses? This variable may impact on whether CBT was received as an inpatient. This is easy to extract from the clinical notes using admission and discharge dates.

Response:

Thank you for highlighting this to us. We agree that the addition of this variable may have served to enhance the research and further elucidate the impact of ethnicity on receipt of CBT in the inpatient context. We have explored whether this information could be extracted, however this has not been feasible due to the demands placed on SLAM ICT systems due to COVID-19. After April the CRIS data moves from an 'opt out' to an 'opt in' system. This means that we would not be able to extract this information for this specific cohort in the future. Consequently, we have acknowledged this as a limitation of the study in the discussion. If we had been able to extract this information we would have included this in an additional post-hoc sensitivity analysis. We have inserted the following: "An additional limitation of this study is we did not extract information regarding the length of inpatient stay. The consequence of this is we do not know the impact of length of stay on the likelihood that someone receives CBT. It is feasible that people who have very short inpatient stays are less likely to receive CBT than those who spend longer in that environment." (p18)

Comment:

Were the authors able to check whether the CBT received as an inpatient was individual or group, as groups tend to be offered on the ward and involve fewer sessions.

Response:

The vast majority of sessions identified across inpatient and outpatient contexts were delivered one-to-one (93.7%; Supplementary Data 2). Service users do tend to receive fewer sessions in an inpatient environment which is supported by the data within Supplementary Data 2 (inpatient mean number of sessions= 2 , IQR=3; outpatient mean number of sessions=7, IQR =15). This study did not seek to differentiate between group and individual sessions in terms of our consideration of whether a person has received CBT or not, or number of sessions.

Discussion:

Comment:

As mentioned, a key limitation is that the study was not able to extract information about the offer of CBT. However, if service users are not accepting CBT, or are dropping out after fewer sessions, then it is important to understand why.

Response:

Thank you for your suggestion. We have added an additional two sentences into the discussion highlighting the importance of trying to understand why service users are not receiving a course of CBT (either not accepting it or not being offered it). We have inserted the following into the manuscript:

“If service users are not accepting CBT or completing a course, or alternatively service providers are not offering or delivering a course of CBT, it is important to understand why. This could be explored in future research.” (pp17-18).

Reviewer: 3

Comment:

In order to adjust for differences in risk profiles and symptom severity effects in the analysis, i was wondering if authors can consider to include interaction effects between ethnicity and other covariates that are significantly related to the outcomes derived from Step 2 analysis. This analysis to ensure the certain group i.e a younger age group, single, male, substance abuser etc within each ethnicity group may have different impact on CBT outcomes?

Response:

Thank you for this suggestion. We have already conducted a number of analyses based on a priori hypotheses and for the purposes of this study have refrained from adding additional post hoc tests for interaction, particularly as tests of interaction are of low power. We have reflected your important point that other factors such as age, gender, marital status and comorbid substance use could be assessed in the future as either mediators (in the association between ethnicity and outcome: access to therapy) or effect modifiers.

Comment:

Have authors perform multiple comparisons test on ethnicity group to examine whether other pair group comparison i.e Irish vs African, Irish vs the Caribbean, Irish vs South Asian, African vs Caribbean, African vs South Asian, Asian vs the Caribbean etc are significant? and whether the significant p values are corrected to control for type 1 error due to multiple comparisons?

Response:

Thank you for this suggestion. We have not performed comparisons between these groups. We determined a priori that the White British group should be the reference group for these analyses. This was based on previous work as well as ongoing as well as current concerns that these ethnic minority groups experience disparities in accessing evidence based treatments, compared with the White British group. As we did not perform these post-hoc analyses it is not necessary to correct for Type 1 errors. The analyses contained within the paper (unless clearly indicated as post-hoc) were

defined a priori and they indicate a consistent pattern (i.e. disparity for Black Caribbean and Black African groups)

Comment:

I am not sure why Area Level Deprivation (IMD) was treated as continuous variable in regression analysis instead of as a categorical variable. Can authors clarify why they decided to decile split into decile instead of other method and keep the original categories in the regression model?

Response:

IMD is not a categorical variable. IMD is a relative measure of deprivation and is calculated by ranking >32,000 LSOAs according to multiple indexes which are used to assess deprivation. Performing a decile split of these data is a typically used technique and is one advocated and utilised by the Department for Communities and Local Government and Office for National Statistics. Please see this link for further information

https://assets.publishing.service.gov.uk/government/uploads/system/uploads/attachment_data/file/464430/English_Index_of_Multiple_Deprivation_2015_-_Guidance.pdf

Comment:

It is not clear about missing data and how the data is handled in the analysis. It would be good if authors can elaborate more in method section.

Response:

We performed complete case analysis . This was inferred (but not very clear) in the exclusion criteria and Figure 1. We have added an additional sentence in making an explicit statement about missing data into the section describing the exclusion criteria: "To this end, only participants with complete data were included." (p8.). We have also provided further details in the results section:

"A total of 5351 cases were excluded due to missing data relating to marital status (n=3678), Index of Multiple Deprivation (n=1308), ethnicity (n=362), gender (n=2) and age (n=1)." (p.12)

.